# Salt-Related Knowledge, Attitudes, and Behaviors on Efate Island, Vanuatu

**DOI:** 10.3390/ijerph16061027

**Published:** 2019-03-21

**Authors:** Emalie Sparks, Katherine Paterson, Joseph Alvin Santos, Kathy Trieu, Nerida Hinge, Len Tarivonda, Wendy Snowdon, Claire Johnson, Jacqui Webster

**Affiliations:** 1The George Institute for Global Health, The University of New South Wales, NSW 2006 Sydney, Australia; jsantos@georgeinstitute.org.au (J.A.S.); ktrieu@georgeinstitute.org.au (K.T.); cjohnson@georgeinstitute.org.au (C.J.); jwebster@georgeinstitute.org.au (J.W.); 2Independent Nutrition Researcher, 2602 Canberra, Australia; katherine.paterson@outlook.com.au; 3Vanuatu Ministry of Health, Iatika Complex, Cornwall St, Port Vila, Vanuatu; itahinge@gmail.com (N.H.); ltarivonda@vanuatu.gov.vu (L.T.); 4Division of Pacific Technical Support, South Pacific Office, World Health Organization, Level 4, Provident Plaza One, Downtown Boulevard, 33 Ellery Street, Suva, Fiji; snowdonw@who.int

**Keywords:** risk factors, health behaviors, salt intake, cardiovascular disease

## Abstract

In Vanuatu, mean salt intake exceeds the recommended maximum daily intake, and contributes to the high proportion of deaths attributable to cardiovascular diseases. Understanding salt-related knowledge, attitudes, and behaviors of the Vanuatu population can inform appropriate interventions. This cross-sectional study was conducted as part of the 2016–2017 Vanuatu Salt Survey. In total, 753 participants aged between 18 and 69 years from rural and urban communities on the Island of Efate were included. Demographic and clinical data were collected and a salt-related knowledge, attitudes, and behaviors survey was administered. Knowledge relating to the need to reduce salt consumption was high, but reported behaviors did not reflect this knowledge. A total of 83% of participants agreed that too much salt could cause health problems, and 86% reported that it was “very important” to lower the amount of salt in the diet. However, more than two-thirds of the population reported always/often adding salt to food during cooking/meal preparation and at the table, and always/often consuming processed foods high in salt. Strategic, targeted, and sustained behavior change programs in parallel with interventions to change the food environment to facilitate healthier choices should be key components of a salt reduction program. Actions should implemented as part of a comprehensive strategy to prevent and control non-communicable diseases in Vanuatu.

## 1. Introduction

Salt, or sodium chloride, is a chemical compound added to food, which can lead to adverse health outcomes when consumed in large quantities [1]. High salt intake increases blood pressure and thereby increases the risk of cardiovascular diseases (CVDs), such as heart disease and stroke [2]. In 2016, CVDs were the leading cause of premature death worldwide, accounting for 32% of all global deaths [3].

Salt reduction is one of the most cost-effective public health interventions [4,5], and named as a World Health Organization (WHO) “best-buy” strategy [6]. Since 2006, the WHO’s target has been to reduce the global average salt intake to less than 5 g/day [7]. Most populations exceed this recommendation, and in fact, the worldwide mean salt intake is estimated to be almost 10 g/day, double the guideline [8,9]. Globally, WHO member states are working towards the global target of a 30% reduction in salt intake by 2025, set at the World Health Assembly in 2013 [10].

The WHO Western Pacific Region has been identified as an area where salt consumption exceeds the WHO guideline, and likely contributes to the high burden of non-communicable diseases (NCDs) [10]. In Pacific Island countries (PICs), there has been a dietary shift away from a traditional diet (consisting primarily of root vegetables, starchy fruit, seafood, and coconut) to a more-Western style diet (comprising packaged foods, refined grains, and meat products) [11], and now foods high in salt are reported to be commonly consumed [12]. Salt intake is also likely to be increasing with the rise in availability of processed foods and commercially prepared meals [13,14]. 

In Vanuatu, it is estimated that 74% of total deaths are attributable to NCDs, with CVDs responsible for 38% of total deaths [3]. In addition, almost 30% of the population was found to have raised blood pressure or was on blood pressure lowering medication in the 2011 WHO STEPwise approach to surveillance of non-communicable disease risk factor (STEPS) survey [15]. Further, mean salt intake has been estimated to be 7.2 g/day from recent spot urine data [16].

Given that salt intake exceeds the WHO recommended maximum in Vanuatu, and the knowledge from other PICs in relation to the dietary shift, it is essential to understand the salt-related knowledge, attitudes and behaviors of the Vanuatu population. This understanding will inform strategic and targeted behavior change interventions, which could be implemented in parallel with interventions to change the food environment, to facilitate healthier choices.

## 2. Materials and Methods

This cross-sectional study was conducted as part of the 2016–2017 Vanuatu Salt Survey, a baseline assessment of salt consumption patterns in the Vanuatu population. Salt intake estimated from spot urine samples has been reported previously [16]. Knowledge, attitudes, and behavior (KAB) were assessed through a questionnaire. The Vanuatu Ministry of Health provided ethical approval for the study. All participants provided written consent.

Participants aged between 18 and 69 years from rural and urban communities on the Island of Efate were surveyed during October 2016 and February 2017. The survey was conducted on Efate Island, as it is the most populous of all the Vanuatu islands and migration from outer islands is high, so the sample may be representative of the country’s population. Participants were divided into two strata (urban and rural) by mapping of enumeration areas (EAs), which was previously undertaken by the Vanuatu National Statistics Office (VNSO) [17]. A total of 28 EAs were randomly selected using the WHO STEPS sampling frame [18], including 14 urban and 14 rural areas.

During the October 2016 data collection period, 50 households were listed in the selected EAs and 27 were randomly selected using the lottery method [19]. Participant selection was undertaken via a convenience sampling method such that eligible household members at home at the time of interviewing were invited to participate in the survey. Only one person from each household was interviewed. Seventeen EAs were sampled in 2016. During the February 2017 data collection, the number of households selected in each of the remaining 11 EAs was increased from 27 to 36, and both households and participants were selected via convenience sampling due to lower than expected participant and household response rates for the urine sampling. Local field researchers were trained to undertake data collection. The survey was administered verbally and recorded on paper. Demographic data collected included age, gender, geographical location, education, and employment. The KAB questionnaire administered was adapted from the WHO STEPS survey (Appendix A) [18]. Three questions assessed the frequency of salt-related behaviors on a 5-point Likert scale, including use of discretionary salt during cooking/meal preparation and at the table, and consumption of high salt processed foods. There were seven binary response questions about what behaviors have been adopted to control salt intake (e.g., avoiding processed foods). Four questions related to knowledge and attitudes, including knowledge of the relationship between salt and health problems (“yes”, “no”), perceived salt consumption (“too little”, “just the right amount”, and “too much”), level of recommended salt consumption (“less than 2 g”, “less than 5 g”, “less than 10 g”), and the importance of lowering salt intake (“not at all important”, “somewhat important”, “very important”). Clinical measures such as body mass index (BMI) and systolic and diastolic blood pressure were collected at the same time.

Data were entered into an excel spreadsheet by the local field researchers at the end of each data collection day, and the team leader checked data quality. The population estimates were made by weighting the sample by age groups (18–44 years or 45–69 years), sex (male or female), and location (rural or urban) distribution of the 2016 mini census population of Efate. The primary outcome was knowledge levels, attitudes and current practices relating to salt, and differences between population subgroups.

Frequency responses to salt use behavior questions from the KAB were aggregated into binary variables: “always/often” and “sometimes/rarely/never”. Subgroup analyses were based on: age (18–44, 45–69 years), gender (male, female), BMI (underweight/healthy weight, overweight/obese), blood pressure (normotensive, hypertensive), geographic location (rural, urban), education (primary school or less, high school or higher) and employment (economically active, not economically active).

Statistical analyses were conducted using STATA/SE (StataCorp LP, College Station, TX, USA) Version 15.0.17. Alpha was set at a *p* <0.05 significance level. Differences between the subgroups were examined using *t*-tests and chi-squared tests. Cohen’s κ tests were run to determine if there was agreement between the three behavior questions and their counterpart salt intake control behaviors.

## 3. Results

Overall, 855 participants were invited to take part in the study. Of these, a total of 753 participants provided consent and completed the KAB (88% response rate). The mean age was approximately 36 years, and 49% of participants were female. Most were urban dwellers (66%) and just over half were economically active (52%). The majority of participants reported completing secondary schooling (45%). Sixty percent of the participants were found to have a BMI in the overweight or obese category and 23% of the population were hypertensive (Table 1).

### 3.1. Knowledge and Attitudes towards Salt Intake

The survey revealed 83% of participants agreed that too much salt could cause health problems, and a similar proportion (86%) reported that it was “very important” to lower the amount of salt in the diet. The majority of participants perceived they consumed either “just the right amount” (44%) or “too much” salt (44%). Only 36% of participants were able to identify that the recommended level of salt consumption was less than 5 g/day (Table 2).

Subgroup analyses revealed that a higher proportion of participants living in urban areas and those with secondary level or higher education agreed that too much salt could cause health problems (urban: 11.7% higher, *p* < 0.001; secondary or higher education: 9.1% higher, *p* = 0.002). Significant differences in perceived salt consumption were found for age (*p* < 0.001) and education level (*p* = 0.002). For perceived recommended maximum salt intake, there were significant differences in the proportions by area (*p* < 0.001) and education (*p* = 0.004; Table 2). There were no subgroups differences in relation to responses on the “importance of lowering salt in the diet”, nor were there any differences found between economic activity status or BMI subgroups for any KAB question (Table 2; Appendix B, Table A1)

### 3.2. Behaviors Relating to Salt Intake

Sixty-seven percent of the population reported always/often adding salt to food at the table, 82% reported always/often using salt during cooking/meal preparation and 68% reported always/often consuming processed foods high in salt. When asked about behaviors performed on a regular basis to control salt intake, 78% of the participants reported that they “avoided processed foods”, closely followed by “not adding salt when cooking” (74%). “Avoiding eating out” (16%) and “looking at sodium labels on food” (31%) were reported the least often as ways to control salt intake (Table 2).

Sub-group analyses demonstrated differences in reported behaviors in relation to gender, age and region. A higher proportion of females reported always/often adding salt to food at the table and while cooking/preparing meals compared to males (*p* = 0.009, *p* = 0.002 respectively). A higher proportion of participants aged 18–44 years and normotensive participants always/often added salt during cooking/meal preparation compared to their counterparts (*p* = 0.007, *p* = 0.008, respectively). A higher proportion of the urban population, younger participants and more highly educated participants reported always/often consuming processed foods high in salt compared to their counterparts (*p* = 0.002, *p* < 0.001, *p* = 0.002, respectively; Table 2). Subgroup differences in reported behaviors to control salt intake are displayed in Table 2 and Table 3.

Cohen’s κ tests revealed there was poor agreement between questions relating to adding salt at the table (κ = 0.036, *p* = 0.124; exact agreement 48.3%), adding salt when cooking (κ = 0.036, *p* = 0.028; exact agreement 37.2%), and consuming processed foods (κ = 0.027, *p* = 0.136; exact agreement 42.8%) [20].

## 4. Discussion

This study demonstrated that knowledge relating to the impact of high salt intake on health and the need to reduce salt consumption was high, however self-reported behaviors did not reflect this knowledge. The majority of the survey population “always/often” added salt at the table and during cooking/meal preparation, and “always/often” consumed processed foods high in salt, illustrating the scope for behavior change interventions to improve salt intake toward the WHO target. The study also highlighted some gaps in knowledge, such as the recommended maximum level of salt intake, demonstrating the need for education and awareness raising activities. 

The results of this survey are comparable with other KAB questionnaires in the Pacific region and globally, showing that people know high salt intakes have negative health impacts, but are not aware of the recommended maximum daily intake or how much salt they are eating (e.g., Samoa [21] and Australia [22,23]). However, compared to the Samoan population, the survey participants in Vanuatu were much more likely to report always/often consuming processed foods high in salt [21]. This is likely because the majority of our sample were from urban areas, which may have greater access to processed foods, whereas most of the Samoan sample were from rural areas [21]. It may also reflect the dietary shift away from the traditional island diets toward a more Western-style diet, high in processed foods [13,14]. The Vanuatu population were also more likely to report always/often adding salt to food whilst cooking, compared to both the Australian and Samoan samples [21,22], which may reflect the lower levels of education in Vanuatu. Whilst there were some differences in behavior, knowledge, and attitudes were similar between the samples. This is particularly remarkable given the difference in country income level [24], as it is known that the majority of dietary salt intake in high-income countries is usually from processed foods, whereas the majority is generally from discretionary salt in low- and middle-income countries [25], and this understanding has implications for behavior change interventions globally.

Some of the findings in our study appear inconsistent and have highlighted the need to review the survey instrument. For example, whilst the majority of participants report “always/often” adding salt at the table and during cooking/meal preparation, and consuming processed foods high in salt, they then list avoidance of these behaviors on a regular basis as strategies to control their salt intake. A possible reason for this inconsistency is differences in the framing of the questions and response options. The initial questions ask about frequency of discretionary salt use on a 5-point Likert scale, allowing a range of responses and, importantly, a middle or neutral point. Contrastingly, the latter asks for a binary yes/no response as to whether behaviors to reduce discretionary salt intake are performed regularly, causing a forced response. Surveys, such as the KAB, are the most popular method to assess knowledge in a variety of fields, including market research and social science [26,27]. Yet, there remains controversy surrounding techniques to measure responses, and there are no conclusive results for the most appropriate answer formats for distinct question constructs [26,27]. A recent systematic review of salt-related KABs revealed a variety of question constructs and answer formats were used, with some using a Likert scale together with binary response or open-ended responses and others using three or more constructs [28]. Interestingly, three studies in different fields—market research, social science, and psychology—indicated that there is no difference in the interpretation of data regardless of answer format, and suggest results are equally reliable between formats [26,29,30]. However, our study suggests that Likert scale and binary responses are not equally reliable in this context. Based on the contrasting data, we recommend further research to strengthen this KAB tool, and validation studies in different populations.

This study adds to the growing body of evidence on food intake, nutrition and health in Vanuatu [16,31], and supports the need for implementation of multi-faceted salt reduction strategies. Salt reduction is one of many nutrition priorities for Vanuatu, and in a resource-constrained environment it is most feasible to address salt consumption patterns as part of a broader strategy to prevent and control NCDs. While mean salt intake in Vanuatu is around 7 g/day [16], which is lower than many other countries and the global average [8,9], the high proportion of the population performing adverse salt-related behaviors is alarming. These results demonstrate the scope for improvement in salt-related behaviors, through strategic, targeted, and sustained behavior change programs, which will be vitally important in reducing population salt intake [32,33]. Behavior change programs should be implemented in parallel with interventions to change the food environment to facilitate healthier choices, including reducing salt in processed foods and meals provided by shops and food outlets.

Subgroup differences detected in this study revealed important considerations for program implementation. People living in rural areas and those with lower levels of education were less likely to understand that too much salt could cause health problems, and fewer still were able to identify the recommended level of salt consumption. With 75% of the Vanuatu population living in rural areas [17], and around 40% having completed primary school or lesser education [34], these results highlight the need to target these subgroups as part of interventions, and develop education materials that take into account the literacy of the population and are culturally appropriate [32]. The education and training of health professionals and school staff will likely be key to achieving behavior change in Vanuatu. To monitor changes in KABs as a result of the program implementation, the core and expanded dietary salt components of the WHO STEPS survey [18] should be included at the next survey.

To further inform future interventions, it is important that more robust dietary assessment methods are undertaken; specifically conducting two 24-h diet recalls on non-consecutive days, to obtain a valid distribution of salt intake and identify major sources of dietary salt [35]. At present, the only available data from the last decade on sources of salt in Vanuatu is from the 2010 Household Income and Expenditure Survey, which showed salt intake was approximately 5 g per day, and major contributors to salt intake were bread, water taro, cabin biscuits, and tinned tuna [29]. However, there are some limitations of using this method to estimate salt intake and identify sources of salt. Specifically, this method makes estimates based on household expenditure data rather than individual consumption data, and does not take into account food waste. This research is a key next step for Vanuatu to guide the actions of researchers, public health workers and policy makers, in implementing multi-faceted salt reduction strategies. 

The strengths of this study included the training and empowerment of local people and the steps taken to ensure cultural sensitivity of questionnaires and procedures. In addition, dissemination of results to the Vanuatu government and population through a written report and media coverage stimulated conversations and raised awareness around salt, further increasing the impact of this work. This was the first study to assess KABs in the Vanuatu population and can be used as a baseline to measure behavior change. There are many challenges to conducting research in PICs, which should be anticipated when conducting future research [36]. Financial and human resource constraints during this study resulted in the administration of the survey occurring at two different time points (October 2016 and February 2017) and only on one island of Vanuatu, Efate Island. Further, the sampling method was modified from random sampling of households in 2016 to convenience sampling in 2017, due to low response rates for the urine collection. However, the estimated intakes from both samples are likely comparable [37]. As with most KAB surveys, the data collected in this study were self-reported, and therefore subject to self-reporter bias. There may be differences in the behaviors reported by participants and actual behaviors which should be considered in the interpretation of these results [28]. In addition, though the WHO STEPs KAB has been widely used, our results suggest the tool may need reviewing and validation in different populations.

## 5. Conclusions

The majority of the Efate population perform adverse health behaviors relating to salt intake, and average salt intake exceeds the WHO recommended maximum. Strategic, targeted, and sustained behavior change programs, in parallel with interventions to change the food environment to facilitate healthier choices, will be vitally important to reducing population salt intake. Incorporating these salt reduction strategies into a comprehensive strategy to prevent and control NCDs, is essential to improving the nutrition status of the Vanuatu population.

## Figures and Tables

**Table 1 ijerph-16-01027-t001:** Weighted sample characteristics.

Characteristics	Overall (*n* = 753)	Female (*n* = 441)	Male (*n* = 312)
Age in years, mean (95% CI)	36.4 (35.5 to 37.3)	35.5 (34.4 to 36.7)	37.3 (35.9 to 38.6)
Female, % (95% CI)	49.2 (45.4 to 52.9)	-	-
Area, % (95% CI)			
Rural	34.1 (30.7 to 37.5)	34.7 (30.5 to 39.2)	33.4 (28.4 to 38.8)
Urban	66.0 (62.5 to 69.3)	65.3 (60.8 to 69.5)	66.6 (61.2 to 71.6)
Completed education, % (95% CI)			
No or minimal formal schooling	13.3 (11.0 to 15.9)	14.5 (11.5 to 18.1)	12.0 (8.9 to 16.1)
Primary level	31.5 (28.2 to 35.1)	33.9 (29.6 to 38.5)	29.2 (24.4 to 34.7)
Secondary level	45.4 (41.7 to 49.2)	42.6 (37.9 to 47.4)	48.2 (42.4 to 53.9)
Tertiary level	9.8 (7.7 to 12.4)	9.0 (6.6 to 12.2)	10.6 (7.4 to 14.8)
Economically active, % ^1^ (95% CI)	52.2 (48.5 to 55.9)	42.7 (38.1 to 47.5)	61.4 (55.6 to 66.9)
Body mass index in kg/m^2^, mean (95% CI)	27.3 (26.9 to 27.7)	28.1 (27.5 to 28.6)	26.5 (25.9 to 27.1)
Overweight or obese, % (95% CI)	59.5 (55.7 to 63.1)	64.0 (59.3 to 68.5)	55.1 (49.3 to 60.8)
Systolic blood pressure in mmHg, mean (95% CI)	121.2 (119.8 to 122.5)	116.8 (115.2 to 118.5)	125.4 (123.3 to 127.5)
Diastolic blood pressure in mmHg, mean (95% CI)	77.9 (77.0 to 78.9)	76.8 (75.7 to 77.8)	79.1 (77.6 to 80.6)
Hypertensive, % ^2^ (95% CI)	22.6 (19.7 to 25.8)	19.2 (15.8 to 23.1)	25.9 (21.3 to 31.2)

^1^ Economically active included: employer, employee, self-employed (not economically active included: unemployed, subsistence living, domestic duties, student, and incapable of working). ^2^ Hypertensive defined as blood pressure ≥90/140 mmHg, or taking hypertensive medication (Normotensive: <140/90 mmHg).

**Table 2 ijerph-16-01027-t002:** Weighted knowledge, attitudes, and behaviors (KAB): Results overall and of subgroup analyses for sex and area.

KAB	Overall (*n* = 753)	Female (*n* = 441)	Male (*n* = 312)	*p*-Value	Rural (*n* = 310)	Urban (*n* = 443)	*p*-Value
Always/often add salt to food, % (95% CI)	66.9 (63.3 to 70.4)	71.6 (67.2 to 75.7)	62.4 (56.6 to 67.8)	0.009	71.0 (65.5 to 76.0)	64.8 (60.0 to 69.3)	0.087
Always/often add salt while cooking, % (95% CI)	81.7 (78.5 to 84.4)	86.2 (82.7 to 89.1)	77.3 (72.1 to 81.7)	0.002	81.3 (76.4 to 85.4)	81.8 (77.7 to 85.3)	0.865
Always/often consume processed foods high in salt, % (95% CI)	68.0 (64.5 to 71.4)	66.6 (62.0 to 70.9)	69.4 (63.9 to 74.4)	0.428	60.7 (54.9 to 66.2)	71.8 (67.3 to 75.9)	0.002
Perceived salt consumption, % (95% CI)							
Too much	43.9 (40.2 to 47.6)	45.1 (40.4 to 49.8)	42.7 (37.1 to 48.5)	0.643	43.4 (37.7 to 49.3)	44.1 (39.4 to 49.0)	0.484
Just the right amount	43.6 (39.9 to 47.4)	41.4 (36.9 to 46.2)	45.7 (40.0 to 51.5)		41.9 (36.3 to 47.8)	44.5 (39.8 to 49.3)	
Too little	8.5 (6.7 to 10.8)	8.9 (6.6 to 11.9)	8.2 (5.6 to 11.9)		9.3 (6.5 to 13.2)	8.1 (5.9 to 11.1)	
Don’t know	4.0 (2.8 to 5.7)	4.6 (3.0 to 7.1)	3.4 (1.8 to 6.2)		5.4 (3.4 to 8.6)	3.3 (1.9 to 5.5)	
Agreed that too much salt could cause health problems, % (95% CI)	83.0 (80.0 to 85.7)	84.5 (80.8 to 87.6)	81.7 (76.8 to 85.7)	0.315	75.3 (69.9 to 80.1)	87.0 (83.4 to 89.9)	<0.001
Perceived recommended salt consumption, % (95% CI)							
Less than 10 g (2 teaspoons)	25.3 (22.2 to 28.7)	25.3 (21.4 to 29.6)	25.3 (20.7 to 30.7)	0.581	30.2 (25.0 to 35.8)	22.8 (19.0 to 27.1)	<0.001
Less than 5 g (1 teaspoon)	36.3 (32.8 to 40.0)	35.2 (30.8 to 39.9)	37.4 (31.9 to 43.1)		27.5 (22.5 to 33.0)	40.9 (36.2 to 45.7)	
Less than 2 g (1/2 teaspoon)	29.0 (25.7 to 32.5)	31.1 (26.9 to 35.7)	27.0 (22.2 to 32.3)		28.4 (23.4 to 33.9)	29.3 (25.2 to 33.8)	
Don’t know	9.4 (7.4 to 11.8)	8.4 (6.2 to 11.3)	10.4 (7.3 to 14.5)		14.0 (10.5 to 18.4)	7.0 (4.8 to 10.1)	
Importance of lowering salt in the diet, % (95% CI)							
Very important	85.7 (82.8 to 88.2)	87.2 (83.7 to 90.0)	84.3 (79.6 to 88.2)	0.385	86.0 (81.3 to 89.7)	85.6 (81.8 to 88.7)	0.996
Somewhat important	7.6 (5.8 to 9.9)	6.1 (4.2 to 8.8)	9.2 (6.3 to 13.1)		7.7 (5.0 to 11.7)	7.6 (5.4 to 10.6)	
Not at all important	2.6 (1.6 to 4.1)	3.1 (1.8 to 5.3)	2.1 (0.9 to 4.8)		2.5 (1.2 to 5.0)	2.7 (1.5 to 4.8)	
Don’t know	4.0 (2.8 to 5.8)	3.7 (2.3 to 5.9)	4.4 (2.5 to 7.5)		3.9 (2.2 to 6.7)	4.1 (2.5 to 6.6)	
Salt intake control (95% CI)							
Avoid processed food	78.2 (75.0 to 81.1)	77.4 (73.1 to 81.1)	79.0 (74.0 to 83.3)	0.596	77.6 (72.5 to 82.0)	78.5 (74.3 to 82.2)	0.773
Look at sodium labels on food	31.1 (27.8 to 34.7)	31.4 (27.2 to 36.0)	30.9 (25.8 to 36.4)	0.870	40.3 (34.7 to 46.2)	26.4 (22.4 to 30.9)	<0.001
Do not add salt on the table	60.9 (57.1 to 64.4)	59.4 (54.6 to 63.9)	62.3 (56.6 to 67.7)	0.428	61.1 (55.2 to 66.6)	60.8 (56.0 to 65.3)	0.935
Buy low-salt alternatives	69.0 (65.4 to 72.3)	65.7 (61.1 to 70.1)	72.1 (66.6 to 77.0)	0.075	73.4 (67.9 to 78.2)	66.7 (62.0 to 71.1)	0.060
Do not add salt when cooking	74.3 (70.9 to 77.4)	72.7 (68.2 to 76.7)	75.8 (70.6 to 80.4)	0.346	74.2 (68.8 to 79.0)	74.3 (69.9 to 78.3)	0.982
Use spices other than salt	60.0 (56.3 to 63.6)	60.7 (56.0 to 65.3)	59.2 (53.5 to 64.8)	0.687	63.0 (57.2 to 68.4)	58.4 (53.6 to 63.1)	0.228
Avoid eating out	15.6 (13.1 to 18.5)	16.8 (13.5 to 20.6)	14.5 (10.9 to 19.1)	0.421	21.8 (17.2 to 27.1)	12.5 (9.7 to 16.1)	0.001

**Table 3 ijerph-16-01027-t003:** Weighted knowledge, attitudes and behaviors (KAB): Results of subgroup analyses for age, education and blood pressure.

KAB	Age 18–44 (*n* = 515)	Age 45–69 (*n* = 238)	*p*-Value	Primary Level or Lower ^1^ (*n* = 363)	Secondary Level or Higher ^2^ (*n* = 389)	*p*-Value	Normotensive ^3^ (*n* = 558)	Hypertensive ^4^ (*n* = 186)	*p*-Value
Always/often add salt to food, % (95% CI)	68.1 (63.7 to 72.1)	63.2 (56.7 to 69.3)	0.208	65.2 (59.9 to 70.2)	68.3 (63.2 to 73.0)	0.397	68.5 (64.3 to 72.4)	63.1 (55.6 to 70.0)	0.196
Always/often add salt while cooking, % (95% CI)	83.7 (80.0 to 86.8)	75.0 (68.9 to 80.2)	0.007	83.3 (79.0 to 86.8)	80.3 (75.8 to 84.2)	0.318	83.9 (80.4 to 86.9)	74.7 (67.6 to 80.7)	0.008
Always/often consume processed foods high in salt, % (95% CI)	71.4 (67.3 to 75.3)	57.1 (50.6 to 63.4)	<0.001	61.8 (56.5 to 66.9)	73.1 (68.3 to 77.4)	0.002	68.6 (64.5 to 72.5)	65.0 (57.6 to 71.6)	0.370
Perceived salt consumption, % (95% CI)									
Too much	47.2 (42.8 to 51.7)	33.1 (27.2 to 39.5)	<0.001	46.0 (40.6 to 51.4)	42.1 (37.1 to 47.3)	0.002	45.3 (41.0 to 49.6)	39.7 (32.5 to 47.3)	0.136
Just the right amount	41.6 (37.2 to 46.1)	50.1 (43.6 to 56.6)		37.2 (32.2 to 42.4)	48.9 (43.7 to 54.1)		44.0 (39.7 to 48.3)	43.9 (36.6 to 51.5)	
Too little	6.9 (5.0 to 9.6)	13.6 (9.8 to 18.6)		11.2 (8.2 to 15.1)	6.4 (4.4 to 9.3)		6.8 (5.0 to 9.3)	12.3 (8.2 to 18.0)	
Don’t know	4.3 (2.8 to 6.4)	3.2 (1.6 to 6.1)		5.7 (3.7 to 8.8)	2.6 (1.4 to 4.9)		3.9 (2.6 to 5.9)	4.1 (1.9 to 8.7)	
Agreed that too much salt could cause health problems, % (95% CI)	82.6 (79.0 to 85.8)	84.4 (79.2 to 88.4)	0.560	78.0 (73.2 to 82.1)	87.1 (83.1 to 90.3)	0.002	82.5 (78.9 to 85.5)	85.9 (79.9 to 90.4)	0.291
Perceived recommended salt consumption, % (95% CI)									
Less than 10 g (2 teaspoons)	25.3 (21.2 to 29.3)	25.5 (20.2 to 31.5)	0.371	23.9 (19.7 to 28.8)	26.3 (22.0 to 31.1)	0.004	25.0 (21.5 to 29.0)	26.5 (20.4 to 33.6)	0.078
Less than 5 g (1 teaspoon)	37.4 (33.2 to 41.9)	32.7 (26.9 to 39.2)		31.2 (26.3 to 36.4)	40.5 (35.6 to 45.7)		38.3 (34.1 to 42.6)	30.5 (23.9 to 37.9)	
Less than 2 g (1/2 teaspoon)	27.6 (23.8 to 31.8)	33.4 (27.6 to 39.9)		32.0 (27.2 to 37.2)	26.6 (22.3 to 31.4)		26.6 (23.0 to 30.6)	35.7 (28.9 to 43.2)	
Don’t know	9.7 (7.4 to 12.7)	8.4 (5.6 to 12.3)		12.9 (9.8 to 16.9)	6.6 (4.3 to 9.8)		10.1 (7.7 to 13.0)	7.3 (4.2 to 12.5)	
Importance of lowering salt in the diet, % (95% CI)									
Very important	85.0 (81.5 to 88.0)	88.0 (83.1 to 91.6)	0.228	85.1 (80.8 to 88.6)	86.2 (82.1 to 89.5)	0.661	85.7 (82.3 to 88.5)	87.9 (81.9 to 92.1)	0.879
Somewhat important	7.7 (6.0 to 10.6)	7.4 (4.6 to 11.7)		7.0 (4.6 to 10.6)	8.2 (5.7 to 11.5)		7.7 (5.6 to 10.5)	7.0 (4.1 to 11.7)	
Not at all important	3.2 (2.0 to 5.2)	0.6 (0.1 to 2.6)		3.1 (1.8 to 5.5)	2.2 (1.0 to 4.6)		2.6 (1.5 to 4.4)	2.4 (0.8 to 6.5)	
Don’t know	4.1 (2.6 to 6.3)	4.0 (2.1 to 7.3)		4.8 (3.0 to 7.5)	3.5 (1.9 to 6.2)		4.1 (2.7 to 6.1)	2.8 (1.0 to 7.6)	
Salt intake control (95% CI)									
Avoid processed food	78.2 (74.3 to 81.7)	78.2 (72.4 to 83.0)	0.990	75.5 (70.6 to 79.8)	80.5 (76.1 to 84.3)	0.111	77.7 (73.8 to 81.1)	80.6 (74.2 to 85.8)	0.405
Look at sodium labels on food	30.8 (26.9 to 35.1)	32.1 (26.4 to 38.4)	0.729	29.5 (24.9 to 34.5)	32.5 (27.9 to 37.6)	0.383	32.7 (28.8 to 36.9)	25.3 (19.4 to 32.3)	0.065
Do not add salt on the table	61.5 (57.1 to 65.7)	58.8 (52.3 to 65.1)	0.503	58.9 (53.5 to 64.1)	62.4 (57.3 to 67.3)	0.340	59.9 (55.6 to 64.1)	62.3 (54.8 to 69.3)	0.580
Buy low-salt alternatives	67.7 (63.4 to 71.7)	73.1 (67.0 to 78.5)	0.143	63.9 (58.6 to 69.0)	73.0 (68.2 to 77.4)	0.011	68.0 (63.8 to 71.9)	72.4 (65.0 to 78.6)	0.291
Do not add salt when cooking	73.9 (69.8 to 77.6)	75.5 (69.5 to 80.7)	0.645	68.2 (63.0 to 73.0)	79.1 (74.6 to 83.0)	0.001	74.3 (70.3 to 77.9)	73.4 (66.1 to 79.5)	0.813
Use spices other than salt	59.6 (55.1 to 63.9)	61.2 (54.6 to 67.4)	0.687	61.7 (56.3 to 66.8)	58.6 (53.4 to 63.6)	0.404	59.2 (54.9 to 63.4)	62.3 (54.8 to 69.3)	0.468
Avoid eating out	16.6 (13.6 to 20.2)	12.5 (8.8 to 17.3)	0.142	16.3 (12.7 to 20.7)	15.1 (11.7 to 19.2)	0.646	16.0 (13.1 to 19.4)	15.3 (10.5 to 21.7)	0.823

^1^ Education of primary school or less: no formal schooling, less than primary school, completed primary school; ^2^ Education of high school or more: completed secondary/high school, completed technical and vocational education and training (TVET) and completed tertiary/university; ^3^ Normotensive: <140/90 mmHg; ^4^ Hypertensive: blood pressure ≥90/140 mmHg, or taking hypertensive medication.

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
