# Peer review of "Salt-Related Knowledge, Attitudes, and Behaviors on Efate Island, Vanuatu"

_ijerph, 2019, doi:10.3390/ijerph16061027_

Round 1
Reviewer 1 Report
This cross sectional survey conducted in Efate Island, Vanuatu described that the knowledge relating to the need to reduce salt consumption was high but reported behaviors did not reflect this knowledge. This study provided useful basic information for the future strategy to promote salt reduction; however, I have several concerns which should be taken into consideration.
First, selection bias is my major concern. Since the response rates were low, I can speculate that only those who were interested in health responded the survey so that knowledge relating to the need to reduce salt consumption was high. If response rates were higher, i.e., those who do not care about their health were included, the rate could be much lower.
Please describe the flow chart illustrating the inclusion of this study. Number of invitation and number of response (response rates) should be clarified.
Page 4 (Table 2), I do not understand why highly educated participants consumed more processed food high in salt as compared with less educated participants.
Author Response
Thank you for your review. Please find detailed response attached.

Reviewer 2 Report
The authors describe a straight-forward survey of dietary salt knowledge, attitudes and behaviors in Vanuatu people. The manuscript is well written and appears acceptable for publication following minor revisions:
Abstract Line 30: " be a key components..."
Materials and Methods: Line 86 and following: briefly describe the methods of data collection and survey administration. The WHO STEPS Instrument is well defined at the reference you cite, but please include 2-3 sentences about questions being verbally asked and also state whether data were recorded on paper or in a computer.
You explain well the need to use robust dietary assessment methods for future salt intake studies. Do any nutrition/health surveillance studies in Vanuatu collect dietary data via FFQ? In future interventions, it will be important for people to understand the amount of salt in foods or in a tsp/Tbsp.
Tables: define the range in the parentheses.
Table in appendix is not cited/named consistently in text and appendix.
Author Response

(The authors gave the same response as above.)

Round 2
Reviewer 1 Report
The authors well responded to my concerns. I have no further comments.